# Parental understanding of our communication of morbidity associated with paediatric cardiac surgery: a qualitative study

Veena Rajagopal [ID],[1] Katherine Brown,[1] Christina Pagel [ID],[1,2] Jo Wray[1]

[1]Heart and Lung, Great Ormond Street Hospital For Children NHS Foundation Trust, London, UK
[2]Clinical Operational Research Unit, University College London, London, UK

**Correspondence to**
Dr Jo Wray; Jo.Wray@gosh.nhs.uk

## ABSTRACT

**Background** Following paediatric cardiac surgery, quality of life may be significantly impacted by morbidities associated with cardiac surgery. Parental understanding of the potential for postoperative morbidity is important for informed decision making. As part of a broader research study, we aimed to elicit parental understanding and experience of the communication of morbidities following their child's cardiac surgery, using traditional focus groups together with an online forum.

**Methods** The Children's Heart Federation set up and moderated a closed, anonymous online discussion group via their Facebook page, focusing on complications, information needs and methods of providing families with information. Additionally, we ran three focus groups with parents/carers, moderated by an experienced independent professional. Focus groups were recorded and transcribed and a single transcript was generated from the online forum. All transcripts were thematically analysed.

**Results** All data were collected in 2014. The forum ran over 3 months in 2014 and involved 72 participants. Focus groups involved 13 participants. Three broad themes were identified: (1) clinicians' use of language, (2) feeling unprepared for complications and (3) information needs of families.

**Conclusions** Clinicians' language is often misunderstood, with wide variability in the way morbidities are described, and between differing teams looking after the same child. Information may not be easily absorbed or retained by families, who often felt unprepared for morbidities that arose after their child's heart surgery. Here, we propose key principles of good communication tailored to the individual receiving it.

### What is known about the subject?

► Stress, time pressure and 'quality' of communication can adversely affect parental understanding, potentially limiting their ability to make informed decisions.

### What this study adds?

► A number of targets for improvement in communication have been identified, based on nine suggested principles of 'good communication'. These principles are:
  – Openness.
  – Avoid jargon.
  – Include emotional and developmental complications during consent.
  – Consistent communication within and across all teams.
  – Appropriate timing of conversations.
  – Tailor conversations to the individual in front of you.
  – Individualise risk—what is likely for this particular child?
  – Provide written and audio-visual material.
  – Provide a liaison link professional with contacts to parental support groups.

## INTRODUCTION

A large majority of children now survive into adulthood following paediatric cardiac surgery,[1][2] making it increasingly important to consider any associated morbidities (as described previously[3]). Furthermore, greater numbers of children with increasingly complex heart disease are now undergoing surgery, resulting in a growing number of children and families living with the impact of a range of morbidities arising as a consequence of this.[4] For those with the most severe morbidities, the impact may be life changing, or in some cases, life-limiting. It is therefore imperative that information about morbidities is communicated in a clear and empathic way.

Very little is known about parental experience and understanding of our communication of morbidity associated with paediatric cardiac surgery. We do know from work in other settings however,[5][6] that levels of distress, time pressure and the quality of 'clinician-parent communication' can adversely affect parental understanding of information given, with the potential to limit parents' ability to make informed decisions.[7] Parents want their

clinicians to be accessible, honest, caring and use lay language at a pace that can be understood.[8]

As part of a wider study looking at the selection, definition, incidence and impact of morbidities following paediatric cardiac surgery,[3 4 9 10] our objective was to elicit parents' perceptions about the way healthcare professionals communicate regarding morbidities associated with paediatric cardiac surgery.

Traditional methods of accessing parental views may exclude certain groups including those who are more geographically remote and those from culturally and ethnically diverse backgrounds.[11 12] Logistics, time and discomfort in face-to-face and group interaction may deter some individuals. Online forums are an important way in which internet users seek healthcare information and communicate with others who have similar healthcare experiences. There is an increasing focus on understanding the potential value of this method of data collection in qualitative research.[13 14] Forums have successfully been used in healthcare research,[15–20] providing a flexible and safe space, where open and honest discussions can be held over a period of months, in a conveniently non-synchronous way for potentially large numbers of users in different geographical areas.[11 12 21] There is no burden on travel, and participation results in minimal disruption to daily life.

By including an online forum as a method of engaging with families, alongside more traditional approaches, we hoped to access the views of those who find it hard to attend in person or take part in interviews and focus groups, adopting a similar approach to that used by our group previously.[22] We report here views elicited from a charity-moderated online forum and from three focus groups.

## METHODS

### Design

A qualitative approach underpinned by an interpretivist framework, employing an online forum in addition to three more traditional focus groups was used as a method of data collection, to access participant viewpoints.

### Participants and data collection

The Children's Heart Federation (CHF), a national parent charity, facilitated and moderated a closed, anonymous, online discussion group via their Facebook page, following a similar approach to that reported in a previous separate study.[22] The discussion group was advertised on the charity's home web page[23] and anyone who had experienced taking care of a child after heart surgery was eligible to participate. Potential participants were directed to the charity's Facebook page where they could access information about the study and governance surrounding it. Anyone wishing to participate was asked to provide basic demographic details (age, gender, ethnicity and geographical region) and on completion of this, they were directed to the closed Facebook

group, where they were able to respond to questions posted there. The research team provided questions to be posted on the forum at the start of the process and the charity were responsible for deciding when new questions should be posted or any prompts introduced, based on participant responses and the rate of responding. New questions were introduced when no new information was being posted—that is, when data saturation had been reached for each question. Questions posted on the forum are shown in box 1. Questions were developed by the research team to address the overall aim of understanding parents' perceptions of how clinicians communicate about morbidities, which morbidities parents think are most important and should be measured and what information they had been given about morbidities prior to their child's surgery. The development of the questions was an iterative process and involved patient representatives who were members of the study team as well as health professionals in order to ensure that content and wording of questions were appropriate. The forum took place over a three month period in 2014.

In addition, we held three focus groups, which took place in Glasgow, Birmingham and London. This additional method of participation was included to enable those who wanted to participate via a more traditional face-to-face approach to do so. These locations were chosen for their broad case mix and ethnic diversity. The focus groups were advertised via the CHF website (as for the online forum)[23] and potential participants contacted the CHF if they wanted to take part. The CHF organised the groups, which were held on a Saturday and were moderated by an independent, experienced researcher using as a framework the same questions that were used for the online forum (see box 1). Focus group participants provided written consent for their participation, recording and use of anonymised quotes in dissemination of the findings. Each focus group was audio-recorded and transcribed verbatim.

Ethical approval was granted by London City Road Research Ethics Committee (study number: 14-LO-1442).

### Data management and analysis

Responses from the online forum were collated into a single transcript. The transcripts from the online forum and focus groups were thematically analysed,[24] enabling identification, analysis and reporting of patterns within the data[25] related to parental understanding and experience of the communication of morbidities following their child's cardiac surgery. Non-parent/carer responses were excluded from analysis. Transcripts were read and codes attached to segments of data independently by members of the research team (JW/CP/VR). Similar codes were then merged to create themes. The researchers met to discuss the themes and agree the descriptive names assigned to them. Discussion continued until consensus was reached.

Participants in the online forum were unknown and not identifiable by the research team; therefore, there

## Box 1 Questions asked on the online forum

1. When thinking about children's heart surgery, what does the word 'complication' mean to you?
2. When thinking about children's heart surgery, what does the word 'morbidity' mean to you? Can you give an example of how this affected you or your child?
3. As the experience of heart surgery becomes more distant and in the part, and your child is older, are there any difficulties that your child has now which you think may relate to the operation?
4. Thinking about yourself and your family, can you help us understand the impact that any complications had on you, your child or any other family members:
   – While your child was still in hospital?
   – After you got home?
5. We are thinking about measuring and recording how often complications happen, but we want to concentrate on the things that matter most to the child and family. Here is a list of problems we know can happen:
   – *Children who have heart problems may sometimes experience different types of brain damage and this may lead to disability ranging from loss of hearing—to problems with learning or movement or all types of disability.*
   – *A child may sometimes need an extra operation during the same stay in hospital that was not planned at the start.*
   – *Infection may sometimes happen after heart surgery.*
   – *A child's kidneys may stop working and need to be supported with a machine for a period of time.*
   – *The muscle that helps a child to breathe may be weakened because the nerve supplying it is bruised or damaged—sometimes an operation is needed to strengthen the breathing muscle (diaphragm muscle).*
   – *A child may need to stay in intensive care for a long time because he/she needed assistance with breathing or he/she needed tubes to remain in place.*
   – *A child's gut may not digest milk or food and he/she may need to have nutrition via a drip.*
   – *A child's heart may become very weak such that a machine to support the body called 'ECMO' was needed.*
   – *Damage can happen to the nerve supply of the heart so that the child needs to have a pace maker put in to regulate the heart beat.*
      – Which of these complications seems the worst or most worrying to you?
      – Could you please tell us why you think one or more of these complications is 'worse' than other complications?
      – Are there things that happened to your child after their operation that you think 'went wrong' which should be on this list but is not?
6. Could you please let us know what information you were provided with regarding the complications of your child's heart surgery?
7. (a) Do you feel you were provided with the right amount of information regarding complications before surgery? (b) How much information would you like (more or less)? How much detail is helpful?
8. What visual (books and so on) methods could be developed for letting families know the risk of complications to better inform them about what might happen after surgery?
9. What do you think about everyone being able to see (on the internet) the numbers of complications occurring after children's heart surgery at different hospitals?

Continued

## Box 1 Continued

10. (a) Do you have any further comments regarding complications following heart surgery? Is there anything we should know to improve services? Is there anything else you can think of? (b) Is there anything we could have done to improve the online study? Do you think we should have asked questions more frequently, for example? Any other suggestions?

was no involvement of participants in transcript review and coding.

### Patient and public involvement

In the broader study, the list of morbidities linked to paediatric cardiac surgery was prioritised by a panel reflecting the views of professionals, parents and patients. This list was used in the questions posted on the forum and at the focus groups.

The reporting of these data is motivated by families reporting inconsistencies in the communication of morbidity related to cardiac surgery.

## RESULTS

### Demographics of participants

#### Online forum

The forum ran over 3 months in 2014 and involved 72 participants (68 mothers, 1 father, 1 patient, 2 grandmothers; age 15–59 years). The vast majority of participants were white British (n=70; 97%) but there was a spread of participants across England, Wales and Scotland.

#### Focus groups

The three focus groups took place in 2014, each lasting approximately 2 hours and in total comprised 13 participants (10 mothers, 2 fathers and 1 adult patient with congenital heart disease). Everyone who expressed an interest in attending a focus group was able to do so, although some were not able to attend on the day due to other commitments or their child being unwell. The ages of the parents' children ranged from 14 months to 24 years (however, only one 'child' was an adult, the others all being 14 years old or younger, with a median age of 5 years at the time of the group). The gender of the participants' children was seven boys and five girls. Two children had recognised syndromes, and one of the mothers was bereaved.

### Collated online forum and focus group data

A number of codes were identified from the online forum and focus group data, and collated to form three themes, with seven subthemes, relating to communication between healthcare professionals and families about complications arising from cardiac surgery. The themes are shown in tables 1–3, with illustrative quotes, and explained in more detail below.

**Table 1** Clinicians' use of language

| Theme: Clinicians' use of language | |
| --- | --- |
| **Sub-themes** | **Quotes** |
| Comprehension | ▶ 'It's really easy to get yourself tied up in knots with risks and percentages, where they're really not explained properly'<br>▶ 'Percentages are rarely accurate anyway, so why do we—as parents—hang on to them so dearly (me included!!!!). We were given 13% but it wasn't properly explained that this was risks (ie, morbidity) not death alone.'<br>▶ 'I think it's assumed that parents have a basic understanding of statistics when it's such a complicated measurement and it's not properly explained to us.'<br>▶ 'It's just jargon…Please…English…so we can understand it!'<br>▶ '…some doctors are better than others…at explaining'<br>▶ 'they go into the usual doctor terminology' 'they use hypoplastic instead of small, stenotic instead of stiff'<br>▶ 'Morbidity to me means death.'<br>▶ 'Just to talk mum to mum…then you're not getting the medical jargon' |
| Consistency | ▶ 'I found a huge discrepancy between the way the cardiologists describe surgery that is, very optimistic, complications are rare, and the surgeons who spell it out in order to cover themselves. Personally, I prefer the latter as it means when it does happen you are aware of it and know it has happened before, whereas the former makes you feel so unlucky and wondering why things have happened.'<br>▶ 'We were given very detailed information by the surgeons on the eve of our sons' op but … up until that time we had only seen cardiologists who were really quite blasé… We were told 'he'll be fine' they do switches all the time. Turns out it was far from that…' |

## THEMES

Figure 1 illustrates the three themes with their related subthemes.

### Clinicians' use of language
#### Comprehension

Language used by clinicians about complications of surgery was often poorly understood by families. Jargon (e.g. words such as 'morbidity', 'stenotic', 'hypoplastic') was used instead of lay language, and percentages were used to communicate risk. These were often confusing to participants and led to misunderstanding.

Some parents felt it would have been useful to hear from another parent who had been through a similar experience before, someone who could tell them 'mum to mum' with 'no jargon', what to expect.

#### Consistency

Participants described different specialties as giving different messages about which morbidities might happen following surgery. While some played down the chance of morbidities happening, others were seen as being more upfront during consent conversations. These inconsistencies between teams led to mistrust, and a perception that some clinicians were being more honest than others.

### Being unprepared for complications
#### Differing priorities of healthcare professionals and families

Parents often felt unprepared for morbidities when they arose. Differing priorities between clinical teams and families may have resulted in some morbidities not being discussed prior to surgery. For example, feeding

**Table 2** Being unprepared for complications

| Theme: Being unprepared for complications | |
| --- | --- |
| **Sub-themes** | **Quotes** |
| Differing priorities of HCP and families | ▶ 'NG feeding was never something I thought about when we considered the prospect of having a congenital heart disease child.'<br>▶ 'We weren't mentally prepared for the longer stay as we were told 'in and out in 5 days'.'<br>▶ 'I really wish someone had prepared me for the psychological side effects… anything explaining how trauma and complications can have a negative impact on your child's self-esteem and mental well-being'<br>▶ 'Tell parents beforehand… This is very likely going to affect development in growth, height, learning and development and things like that'<br>▶ 'They do tell you some of the physical things that might happen' but not 'how it might affect… a person's behaviour or emotions' |
| Timing of consent | ▶ 'You know when they do the consent forms it's usually the night before surgery which… is a bad idea, because you're not taking that in… if you did it a week before… you take in a lot more… it's easy for you to digest and understand' |

HCP, healthcare professionals.

**Table 3** Information needs of families

| Theme: Information needs of families | |
| --- | --- |
| Subthemes | Quotes |
| No right amount of information | ▶ *'Some parents will want to know everything and others want to know as little as possible'*<br>▶ *'If we knew all the potential outcomes I think signing the consent would have been so much harder'*<br>▶ *'You can't have a blanket rule of 'We must tell them every possible thing that could go wrong' or, 'We only tell them the most common'. You need to look at it case-by-case'*<br>▶ *'It's such a delicate balance… you've got to try and test the waters with the patient's parents and the patient's themselves to find out how much information… you need to give this person. It has to be an individual case-by-case scenario'* |
| Types of information | ▶ *'Something to take away and look at and digest in your own time'*<br>▶ *'Even a simple information sheet… you might not have internet access… you might need to go into your room or sit by the bed and have another look at it a bit later'*<br>▶ *'Little Hearts Matter… DVD pack for antenatal diagnosis. It was brilliant.'* |
| Access to staff/ lay support | ▶ *'Community Liaison Nurses are very useful and parents should be given contact numbers as a matter of course'*<br>▶ *'knowing who to ask… that there's somebody you can ask about things you might spot… signposting'* |

difficulties, and the need for a nasogastric tube were of huge significance to families; however, parents were frequently not made aware of this as a potential morbidity, perhaps because clinicians did not consider it with the same significance.

Another important example identified by many parents was the unanticipated psychological side effects on their child, the siblings and themselves, which were invariably lacking from discussions prior to cardiac surgery.

The lack of warning, and a consequent feeling of being completely unprepared to deal with these morbidities once they arose, was a clear difficulty expressed by parents.

### Timing of consent

The timing of consent conversations was also highlighted as being suboptimal. Examples were given of consent being taken the night before surgery, when parents were feeling anxious and left with little time to reflect on or think through what had been said.

Parents identified that this affected their ability to understand and recall what they were told about the risks of surgery, contributing to making a difficult time more challenging.

### Information needs of families

#### There is no right amount of information

It was clear that there was no 'right amount of information' that suits every individual. While some parents wanted to know about all possibilities, others felt that too much information would have been overwhelming, potentially paralysing them in the decision-making process.

Participants felt that the detail of information being conveyed should be tailored according to the needs of the individual receiving it, rather than using a 'one size fits all' approach.

#### Types of information

Written and audio-visual material were reported as helpful supplements. Being able to take something back

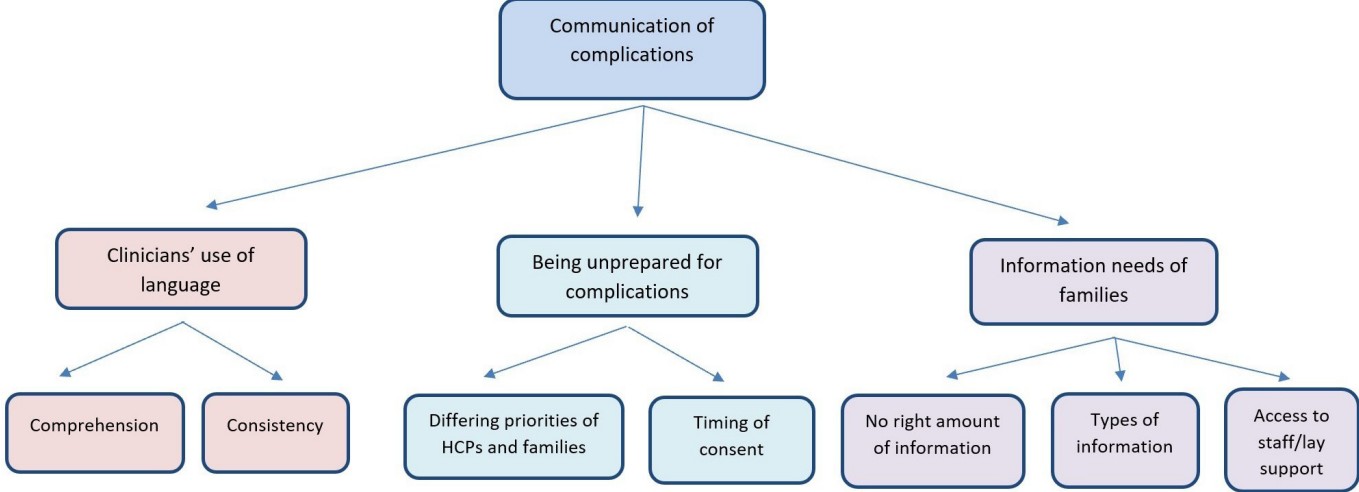

**Figure 1** Summary of feedback from parents about how complications were communicated to them by HCPs. HCPs, healthcare professionals.

to their accommodation, that could be read and digested in their own time, was valued. As well as more high-tech information, simple leaflets were appreciated as something that could be read without the need for internet access and a phone/computer.

## Access to staff/lay support

Parents valued access to a member of staff they could contact with further questions, and who could signpost them to other information as needed.

In addition, parents expressed that being able to speak to another parent, who had gone through something similar, and who *actually knew* how it felt to be a parent going through this, would have been a source of comfort and help.

## DISCUSSION

Healthcare communication is a vital skill, influencing understanding of information and hence informed decision making. The importance of 'good communication' however, goes beyond just understanding; it has been suggested that the way in which clinicians communicate may also impact on the psychological adjustment and functioning of parents and families.[8] It is widely acknowledged that paediatric cardiac surgery is very stressful for parents, particularly in view of the risk of morbidities associated with the operation, and how clinicians communicate this is likely to have an impact on parents. We therefore wanted to explore parents' perceptions of how clinicians communicate about morbidities and what information they had been given about morbidities prior to their child's surgery. The feedback we obtained suggests that there is wide variability and lack of consistency in the way clinicians describe morbidities, much of which may not be absorbed or retained by parents, particularly during times of extreme stress and distress.[26–28] A recent survey exploring communication between parents and clinical teams following children's heart surgery[28] supports our findings with reports of inconsistent communication (almost two-thirds of parents reporting this), particularly where complications arose or children were in hospital for longer periods of time.

Our study describes clinician, parental (predominantly maternal) and situational factors that may influence parental understanding of potential morbidities following paediatric cardiac surgery, representing areas that can be targeted to improve this (see box 2):

► Clinicians use of jargon, the lack of consistency between clinicians and an individual clinician's skill in communication have been previously identified as important factors in patient comprehension and adherence[29 30] and in the current study all of these aspects impacted reported parental satisfaction and understanding of the conversation. Individual communication skills vary, with one participant noting that some clinicians are 'just better' at it than others. Though this may be an innate strength or weakness

---

**Box 2  Principles of good communication**

1. Openness.
2. Avoid jargon.
3. Include emotional and developmental complications during consent.
4. Consistent communication within and across all teams.
5. Appropriate timing of conversations.
6. Tailor conversations to the individual in front of you.
7. Individualise risk—what is likely for *this* particular child?
8. Provide written and audio-visual material.
9. Provide a liaison link professional with contacts to parental support groups.

---

within an individual, clinicians need to reflect and improve on their own communication skills, in the same way they would for any technical or procedural skills required of them. Observation of exemplar mentors and simulation training in communication including discussions with parents of children who have undergone cardiac surgery are useful strategies to improve these skills and gain parent centred perspectives.[26]

► We know that the amount of information given by clinicians can have both positive and negative effects on parental anxiety.[7 31] This was also reflected in our findings, suggesting that information given to parents needs to be tailored to the information needs of the individual receiving it. Taking the time to know what is right for the parent a clinician is speaking to is an important investment in the parent's understanding and in their ability to make decisions and cope with their child's postoperative course. In addition, parents want to know what is likely to happen to *their* child, such that the risk of morbidities would ideally be tailored to the individual characteristics of the child for whom surgery is being discussed. However, as the population of cardiac children become increasingly complex, and as surgical and medical techniques evolve, tailoring assessments of risk is difficult and always subject to some uncertainty. How clinicians support parents and families in coping with uncertainty needs to be an important part of specialist care.

► Stress and time pressure are barriers to informed consent.[32 33] Participants commented on the difficulties they faced with consent the night before surgery. Where possible, a staged consent approach[34] should be used, such that an initial consultation will be used just to relay information, followed by a period of time to allow consideration by the parents, before a second conversation where consent is actually sought.

Many of our participants commented on the feeling of being 'unprepared'. We suggest future work should seek to understand if clinicians are able to mitigate some of the short and long-term distress experienced by families in this situation, through better communication and understanding of the parental perspective of morbidities and their impact on the child and family.

Parents described the importance of other sources of information that they could access whenever they wanted to, such as written information and internet resources. Written information has been found to result in significant decreases in parental anxiety and improved parental comprehension and satisfaction prior to their child's surgery.[35] Multimedia-based health information has also been found to reduce parental anxiety[36] although evidence supporting the greater effectiveness of any one method of information provision is lacking,[36 37] highlighting the importance of individually tailoring information provision. Parents in our study also valued the support offered by staff and in particular the support of other parents who had been through similar experiences, replicating findings with other illness groups.[38] Ensuring the availability of health professionals such as psychologists and medical social workers to provide ongoing support to families could help alleviate parental stress and provide an opportunity for information to be repeated and further discussed.

In this study, the online forum gave us access to many more parents than the focus groups (72 online, 13 at focus groups). This correlates with other reports suggesting the relative ease of online discussion compared with attending in person focus groups.[11 12 21]

There are a number of limitations which need to be taken into account when interpreting the findings. First, the data we obtained relied on participants' retrospective recall of conversations (with its inherent limitations on accuracy). Future work to look at clinician–parent communication in real time would provide a valuable insight into what is *actually* said, and the interpretation at the time—of parents and clinicians—that is, what parents understood from the conversation and what clinicians believe the parents understood from the conversation. Furthermore, using an online forum to collect parent perceptions does not result in the same depth of information that would be possible in individual interviews and the approach precludes probing for further detail.

We recognise that both the online forum and the focus groups can in themselves exclude certain individuals due to lack of familiarity, language barriers, literacy barriers and lack of internet resources. In addition, our study lacked the views of males and non-Caucasians, who were under-represented in our participant sample, despite the chosen locations for the focus groups having a broad ethnic and socioeconomic mix. The challenges of including ethnically, culturally and linguistically diverse populations, as well as fathers, in paediatric research studies are well documented.[39 40] In order to capture these important often unheard views, we must find and adopt an innovative approach that successfully includes minority groups to ensure the broadest capture of parental/family views. Despite specifically choosing a data collection method to increase the accessibility of the research to potential participants, the fact that our participants did not reflect a broad range of ethnic groups or gender also limits the transferability of our findings to the wider population of parents of children with congenital heart disease in the UK.

Finally, we did not collect the same demographic information from participants in the online forum and the focus groups, which limits our ability to describe certain aspects of our participants across both data collection approaches.

## CONCLUSION AND SUGGESTIONS FOR FUTURE PRACTICE

Our findings indicate the need for an individualised approach to communication about morbidity associated with paediatric cardiac surgery, based on 'high quality conversations', in which certain *key principles* of good communication are followed, as discussed above. This is not unique to paediatric cardiac surgery—rather it is relevant in any situation in which an individual is undergoing a medical or surgical intervention. Future work to investigate the impact of 'good communication' on the short-term and long-term psychological morbidity of parents and families would be a valuable next step, together with identifying how this may impact on physical and psychological outcomes in children undergoing cardiac surgery.

**Acknowledgements** Research at Great Ormond Street Hospital is supported by the NIHR GOSH BRC

**Funding** This project was funded by the National Institute for Health Research Health Services and Delivery Research programme (Project No: 12/5005/06).

**Competing interests** None declared.

**Patient consent for publication** Not required.

**Provenance and peer review** Not commissioned; externally peer reviewed.

**Data availability statement** All data relevant to the study are included in the article or uploaded as supplementary information. Note: The original uncoded transcripts from the online forum and three focus groups are under the care of Dr Wray. Email: jo.wray@gosh.nhs.uk.

**ORCID iDs**
Veena Rajagopal http://orcid.org/0000-0002-6946-8272
Christina Pagel http://orcid.org/0000-0002-2857-1628

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
