## [Reviewer comments · BMJ Paediatrics Open]

ARTICLE DETAILS

TITLE (PROVISIONAL)	Parental understanding of our communication of morbidity associated with paediatric cardiac surgery: a qualitative study.
AUTHORS	Rajagopal, Veena; Brown, Katherine; Pagel, Christina; Wray, Jo

VERSION 1 – REVIEW

REVIEWER	Reviewer name: Manning, Joseph Institution and Country: Nottingham University Hospitals NHS Trust, Nottingham Children's Hospital Competing interests: I have no conflict of interest. I do not work with the team or any of its members.
REVIEW RETURNED	09-Nov-2018

GENERAL COMMENTS	Thank you for inviting me to review this paper which reports a qualitative study which sought the perspectives of parents in relation to the communication of morbidities associated with paediatric cardiac surgery. Despite data being collected over 4 years ago, the focus of the paper appears to be an under-researched topic and findings appear to have contemporary relevance to clinical practice as well as future research. From my review there are some areas of the paper that i would encourage the authors to revise to strengthen the paper. These include: Title 1. This needs to include the term 'communication' as this is the focus of the paper Abstract 1. Background section needs revising to be more focused. Substantiate statement in Ln 13.2. Results section need to clarify if the focus groups were conducted in 2014. Introduction 1. It is unclear from the introduction what is the existing understanding of parental experiences of communication in relation to paediatric [cardiac] surgery. Whilst i realise that you go on to discuss your findings in this literature in the discussion, it would be useful to present a summary of this information to situate the reader. Furthermore, this paper might be useful to include: Levetown, M., 2008. Communicating with children and families: from everyday interactions to skill in conveying distressing information. Pediatrics, 121(5), pp.e1441-e1460.2. Please specify what morbidities children can experience post-cardiac surgery.3. Page 3, Ln 35-37, please substantiate claim.4. Page 3, Ln 50- Page 4, Ln 19 is rationale for methods as opposed to introduction and therefore should be moved into methods.
--

4. Please define 'hard to reach groups' .

Methods

1. There is no reference to methodological orientation or theoretical (epistemological/ontological) position. Please can you expound this clearly in the text.

2. Page 4, Ln 54-56, you make reference to another study that team have conducted about parental experiences of discharge. It is not clear if both studies used the data from the same forum or whether the forum was set up specifically for this study and the data generated has not been used elsewhere. Could you please clarify this point.

3. Page 5. Unclear of the rationale for conducting focus groups in addition to the online forum, the year when the focus groups were held, sampling approach adopted, and any eligibility criteria for inclusion in the focus groups (e.g. parent). Please explicate this in the methods.

4. Page 6. Please include a sentence that provides rationale for selecting this type of analysis and why it was congruent with your research aim.

5. It would be useful to include a sentence on whether participants were involved in reviewing the transcripts/coding, and rationale provided if this was not undertaken.

Results

1. Demographics of participants- this would be easier to read if summarised in a table for both the online forum and the focus groups.

2. In both the online forum and the focus groups you include patients (n=2). Could you please narrate in the text how you managed that data (e.g. excluded it from the analysis) as the aim of your study was about parental experiences and understanding.

3. Did you reach data saturation? if so please state this in the results.

4. Page 9, Ln 11-18, It is unclear how this theme relates to communication/understanding of morbidities. Please can you revise to make more explicitly linked to the aim of the study/paper.

Discussion

1. Page 10, Ln 25-34, I think this paragraph would be better situated in the introduction section- please move.

2. The discussion is centred on the clinicians role in communicating and the psychological impact. Whilst this is interesting it would be useful to also integrate some literature pertaining to information sources, and peer support as these were identified in the findings.

3. Page 11, Ln 31, remove idiom.

4. The main rationale for employing online methods to explore this phenomena is to access 'hard to reach' populations. It would be useful to evaluate here if you have achieved that and what impact that has had.

Limitations

1. Ln 26- need to expound the limitation (e.g. as this could affect recall?)

2. Ln33-35, Unsure why there is a question there? Please revise or remove.

3. Ln 43- Please ensure when you make your revisions that there is congruence between your theoretical position and your knowledge claims (e.g. generalisability).

	I would expect your findings to be transferable to reflect the characteristics of your sample as opposed to generalisable- please revise accordingly. 4. Another limitation that is not listed is the method in which your recruited participants for both the forum and focus group. This was via a website and therefore could exclude participation by non-english speakers, those illiterate (computer and in English language) etc. This needs to be acknowledged in your limitations. Conclusion 1. Ln 13-30 should be moved to limitations section. 2. Overall the conclusion could be strengthened by (a) linking the findings to the aim of the paper/study; (b) expounding the novel elements of the findings, (c) clear recommendations for research and practice. This needs to be specific but also useful for the broad readership of ADC. 3. I am unclear of how informative/useful figure 2 is. Currently it reads as an add on and could be interpreted as patronising (e.g. 'Honesty'). I would recommend revising it so it is more informative/detailed or signposting to established frameworks - some of which have been included in this review : King, A. and Hoppe, R.B., 2013. "Best practice" for patient-centered communication: a narrative review. Journal of graduate medical education, 5(3), pp.385-393.. I hope you find these points useful in revising your manuscript.
--	---

REVIEWER	Reviewer name: Lotto, Attilio Institution and Country: Alder Hey Children's, Cardiac Surgery Competing interests: None
REVIEW RETURNED	20-Nov-2018

GENERAL COMMENTS	Dear Authors I have enjoyed reading your paper. In summary you present the results of a qualitative research the carried out 4 years ago on parental perception and understanding of morbidity associated with congenital cardiac surgery. You run an online forum via the Children Heart Federation and 3 focus groups in Glasgow , Birmingham and London. You have interviewed parents and used a qualitative research tools to analyse their responses, grouping them in 3 main themes: Clinicians' use of language, Feeling unprepared for complications , Information needs for families. The paper reads well, but there are few aspects in the methods and in the presentation of the results that merit consideration. Methods: despite running focus groups in Birmingham, there was very little engagement from other ethnic groups than white British. Despite acknowledging this as a limitation, little is provided to explain why this has happened, and maybe worth try to comment on your methods to try to including all ethnicities. Also the sample is represented only by mothers, hence a very unbalanced group. A comment on that might be useful during the discussion so to remember the reader that the views are those of mothers of children undergoing CHD surgery. In the findings section, the comments from the parents are not included in the body of the paper, but in an appendix. I would suggest that, for a qualitative paper, it might be useful extending the world count to allow some of the comments to be included into the main text. You might want to negotiate this with the Editor. I think it is an interesting paper, and with those adjustments, I think it touches on a topic not very well investigated, so far.
--

VERSION 1 – AUTHOR RESPONSE

Reviewer: 1

Comments to the Author

Thank you for inviting me to review this paper which reports a qualitative study which sought the perspectives of parents in relation to the communication of morbidities associated with paediatric cardiac surgery. Despite data being collected over 4 years ago, the focus of the paper appears to be an under-researched topic and findings appear to have contemporary relevance to clinical practice as well as future research.

From my review there are some areas of the paper that i would encourage the authors to revise to strengthen the paper. These include:

Title

1. This needs to include the term 'communication' as this is the focus of the paper

RESPONSE: The title has been revised and now reads: "Parental understanding of our communication of morbidity associated with paediatric cardiac surgery: a qualitative study".

Abstract

1. Background section needs revising to be more focused. Substantiate statement in Ln 13.

RESPONSE: Thank you. We have revised the background section of the abstract to be more focused.

2. Results section need to clarify if the focus groups were conducted in 2014.

RESPONSE: We have added in an additional statement to make it clear that all data collection took place in 2014.

Introduction

1. It is unclear from the introduction what is the existing understanding of parental experiences of communication in relation to paediatric [cardiac] surgery. Whilst i realise that you go on to discuss your findings in this literature in the discussion, it would be useful to present a summary of this information to situate the reader. Furthermore, this paper might be useful to include: Levetown, M., 2008. Communicating with children and families: from everyday interactions to skill in conveying distressing information. *Pediatrics*, 121(5), pp.e1441-e1460.

RESPONSE: Thank you for these suggestions. We have revised the introduction to include the following paragraph to situate the reader: "Very little is known about parental experience and understanding of our communication of morbidity associated with paediatric cardiac surgery. We do know from work done in other settings, however [5,6], that levels of distress, time pressure and the quality of 'clinician-parent communication' can adversely affect parental understanding of information given, with the potential to limit parents' ability to make informed decisions. [7] Parents want their clinicians to be accessible, honest, caring, and use lay language at a pace that can be understood. [8]"

Thank you for the Levetown reference – this has been a useful addition and cited in the revised manuscript as reference 28.

2. Please specify what morbidities children can experience post-cardiac surgery.

RESPONSE: We have added in a citation to previous work which defines important morbidities associated with paediatric cardiac surgery (reference 3).

3. Page 3, Ln 35-37, please substantiate claim.

RESPONSE: Reference 4 “Incidence and risk factors for important early morbidities associated with pediatric cardiac surgery in a UK population” was added to substantiate the sentence ‘As more children with increasingly complex heart disease successfully undergo surgery, there are a growing number living with the impact of a range of morbidities arising as a consequence of surgery’ (previously lines 35-37 on page 3).

4. Page 3, Ln 50- Page 4, Ln 19 is rationale for methods as opposed to introduction and therefore should be moved into methods.

RESPONSE: Thank you for this suggestion. We have left the text in the introduction rather than moving it into the methods as this is providing the reader with background about how parental views have been accessed previously. Whilst it is also providing a rationale for our final choice of methods, we think that it should remain as background material for our study.

5. Please define 'hard to reach groups'.

RESPONSE: We have revised this sentence and added further clarification about what we mean by 'hard to reach groups'.

Methods

1. There is no reference to methodological orientation or theoretical (epistemological/ontological) position. Please can you expound this clearly in the text.

RESPONSE: We have added the following text: “A qualitative approach underpinned by an interpretivist framework, employing an online forum in addition to three more traditional focus groups was used as a method of data collection, to access participant viewpoints.”

2. Page 4, Ln 54-56, you make reference to another study that team have conducted about parental experiences of discharge. It is not clear if both studies used the data from the same forum or whether the forum was set up specifically for this study and the data generated has not been used elsewhere. Could you please clarify this point.

RESPONSE: These are two separate online forums for two different studies, the earlier one of which has been published. The reference is present just to highlight experience using an online forum. We have highlighted that this reference is related to a separate study in the manuscript: “following a similar approach to that reported in a previous separate study. [22]”

3. Page 5. Unclear of the rationale for conducting focus groups in addition to the online forum, the year when the focus groups were held, sampling approach adopted, and any eligibility criteria for inclusion in the focus groups (e.g. parent). Please explicate this in the methods.

RESPONSE:

Rationale for conducting focus groups in addition to the online forum: using focus groups in addition to the online forum provided a further method by which parents/carers could participate and enabled those who wanted to contribute via a more traditional face to face method to do so. We have added text to that effect.

Year when the focus groups were held: This has been clarified in the text; the focus groups and the online forum both took place in 2014.

Sampling approach adopted and eligibility criteria for inclusion: We have included a link to the online advertisement for the focus groups and online forum which demonstrates the sampling approach and gives detail of the eligibility criteria (see reference 23).

4. Page 6. Please include a sentence that provides rationale for selecting this type of analysis and why it was congruent with your research aim.

RESPONSE: We have added further text explaining the rationale for the choice of analysis and how this was congruent with our research aim.

5. It would be useful to include a sentence on whether participants were involved in reviewing the transcripts/coding, and rationale provided if this was not undertaken.

RESPONSE: We have added the following sentence to address this point: "Participants in the online forum were unknown and not identifiable by the research team, therefore there was no involvement of participants in transcript review and coding."

Results

1. Demographics of participants- this would be easier to read if summarised in a table for both the online forum and the focus groups.

RESPONSE: Unfortunately, we do not have the same demographic data collected from both the forum and focus groups. We do not think that a table is the best way to display the demographics that we do have, and have therefore left this as text in the paper.

2. In both the online forum and the focus groups you include patients (n=2). Could you please narrate in the text how you managed that data (e.g. excluded it from the analysis) as the aim of your study was about parental experiences and understanding.

RESPONSE: We did exclude responses that were not from parents/carers and have added a comment in the paper: "The transcripts from the online forum and focus groups were thematically analysed (non -parent/carer responses were excluded from analysis)."

3. Did you reach data saturation? if so please state this in the results.

RESPONSE: We have added further text to the effect that new questions were introduced when no new information was being posted – i.e. when data saturation had been reached for each question. We have added this to the methods section as it reflects the approach the charity took to posting the questions.

4. Page 9, Ln 11-18, It is unclear how this theme relates to communication/understanding of morbidities. Please can you revise to make more explicitly linked to the aim of the study/paper.

RESPONSE: We have added some further text to make the link between this theme and parents' understanding of morbidities clearer: "Parents identified that this affected their ability to understand and recall what they were told about the risks of surgery, contributing to making a difficult time more challenging."

Discussion

1. Page 10, Ln 25-34, I think this paragraph would be better situated in the introduction section please move.

RESPONSE: We have moved this paragraph to the introduction.

2. The discussion is centred on the clinicians' role in communicating and the psychological impact. Whilst this is interesting it would be useful to also integrate some literature pertaining to information sources, and peer support as these were identified in the findings.

RESPONSE: Thank you for this suggestion. We have added a further paragraph about information sources and peer support.

3. Page 11, Ln 31, remove idiom.

RESPONSE: This has been removed.

5. The main rationale for employing online methods to explore this phenomenon is to access 'hard to reach' populations. It would be useful to evaluate here if you have achieved that and what impact that has had.

RESPONSE: This has been amended and addressed in the Limitations paragraph: "In addition, our study lacked the views of males and non-Caucasians, who were under-represented in our participant sample, despite the chosen locations for the focus groups having a broad ethnic and socio-economic mix. The challenges of including ethnically, culturally and linguistically diverse populations, as well as fathers, in paediatric research studies are well documented. [37, 38] In order to capture these important often unheard views, we must find and adopt an innovative approach that successfully includes minority groups to ensure the broadest capture of parental/family views."

Limitations

1. Ln 26- need to expound the limitation (e.g. as this could affect recall?)

RESPONSE: We have clarified this with the comment "with its inherent limitations on accuracy" to highlight that a problem with retrospective recall is that it relies on past memory of what one thought was said/done rather than being an accurate description of what was actually said or done.

2. Ln33-35, Unsure why there is a question there? Please revise or remove.

RESPONSE: We have reworded this to remove the question.

3. Ln 43- Please ensure when you make your revisions that there is congruence between your theoretical position and your knowledge claims (e.g. generalisability). I would expect your findings to be transferable to reflect the characteristics of your sample as opposed to generalisable- please revise accordingly.

RESPONSE: Thank you. We have revised the text as suggested.

4. Another limitation that is not listed is the method in which your recruited participants for both the forum and focus group. This was via a website and therefore could exclude participation by non-english speakers, those illiterate (computer and in English language) etc. This needs to be acknowledged in your limitations.

RESPONSE: Thank you. We have amended this and this limitation is now addressed in paragraph two of the limitations section: "We recognise that both the online forum and the focus groups can in themselves exclude certain individuals due to lack of familiarity, language barriers, literacy barriers and lack of internet resources".

Conclusion

1. Ln 13-30 should be moved to limitations section.

RESPONSE: This has been moved to the limitations section.

2. Overall the conclusion could be strengthened by (a) linking the findings to the aim of the paper/study; (b) expounding the novel elements of the findings, (c) clear recommendations for research and practice. This needs to be specific but also useful for the broad readership of ADC.

RESPONSE:

We have highlighted the data collection methods (online forum) as a novel method of collecting data to address our aim.

We have put recommendations for practice in the discussion paragraph (e.g. Observation of exemplar mentors and simulation training in communication, including discussions with parents of children who have undergone cardiac surgery, are useful strategies to improve these skills and gain parent centred perspectives. [28]) and also in Table 3. We have also added further text to the conclusion paragraph about the wider relevance of our findings.

3. I am unclear of how informative/useful figure 2 is. Currently it reads as an add on and could be interpreted as patronising (e.g. 'Honesty'). I would recommend revising it so it is more informative/detailed or signposting to established frameworks - some of which have been included in this review : King, A. and Hoppe, R.B., 2013. "Best practice" for patient-centered communication: a narrative review. Journal of graduate medical education, 5(3), pp.385-393..

RESPONSE: We changed the word "Honesty" to "Openness" to avoid offending readers. The principle of being Open/Honest however is vital in good communication, and although it may seem at first obvious, it is a common problem that clinicians are not open and honest in their discussions – this is a common finding in our experience. It is not because they are being dishonest, it is perhaps more that they are trying to be optimistic and not frighten families. However, the results from this paper tell us that mothers want us to be honest with them, however challenging this may be for them to hear.

Thank you for the King & Hoppe reference. We have included it, but think that the more detailed tables (that for example are included in the King paper), are not what we wanted here. The purpose of figure 2 (now table 2), is to be a brief bullet pointed aide memoir when preparing for an important communication (e.g. consent) with families.

I hope you find these points useful in revising your manuscript.

Thank you for your constructive feedback.

Reviewer: 2

Comments to the Author

Dear Authors I have enjoyed reading your paper.

In summary you present the results of a qualitative research the carried out 4 years ago on parental perception and understanding of morbidity associated with congenital cardiac surgery. You run an online forum via the Children Heart Federation and 3 focus groups in Glasgow , Birmingham and London. You have interviewed parents and used a qualitative research tools to analyse their responses, grouping them in 3 main themes: Clinicians' use of language, Feeling unprepared for complications , Information needs for families.

The paper reads well, but there are few aspects in the methods and in the presentation of the results that merit consideration.

Methods: despite running focus groups in Birmingham, there was very little engagement from other ethnic groups than white British. Despite acknowledging this as a limitation, little is provided to explain why this has happened, and maybe worth try to comment on your methods to try to including all ethnicities. Also the sample is represented only by mothers, hence a very unbalanced group. A comment on that might be useful during the discussion so to remember the reader that the views are those of mothers of children undergoing CHD surgery.

RESPONSE: Thank you for these suggestions. We have expanded the comments about lack of engagement from minority groups/fathers: "In addition, our study lacked the views of males and non-Caucasians, who were under-represented in our participant sample, despite the chosen locations for the focus groups having a broad ethnic and socio-economic mix. The challenges of including ethnically, culturally and linguistically diverse populations, as well as fathers, in paediatric research studies are well documented. [37, 38] In order to capture these important often unheard views, we must find and adopt an innovative approach that successfully includes minority groups to ensure the broadest capture of parental/family views".

Comment about mothers included: "Our study describes clinician, parental (predominantly maternal) and situational factors that may influence..."

In the findings section, the comments from the parents are not included in the body of the paper, but in an appendix. I would suggest that, for a qualitative paper, it might be useful extending the word count to allow some of the comments to be included into the main text. You might want to negotiate this with the Editor.

RESPONSE: We would be glad to negotiate this with the Editor should our paper be accepted for publication.

I think it is an interesting paper, and with those adjustments, I think it touches on a topic not very well investigated, so far.

Thank you for this positive feedback.

VERSION 2 – REVIEW

REVIEWER	Reviewer name: Attilio Lotto Institution and Country: Alder Hey Children's Liverpool John Moores University Liverpool United Kingdom Competing interests: None
REVIEW RETURNED	02-Dec-2019
GENERAL COMMENTS	Dear Authors, I have read the revised version of this paper. You have made all the required corrections from both reviewers. I still think it is worth asking for an extension on the word counting to be able to include into the body of the paper some of the quotations from parents. I am now content with the paper and I think it is ready for publication.
REVIEWER	Reviewer name: Dr Shine Kumar Institution and Country: Clinical Associate Professor, Dept. of Pediatric Cardiology, Amrita Institute of Medical Sciences, Amrita Vishwa Vidyapeetham, Kochi, Kerala, India

	Competing interests :None
REVIEW RETURNED	14-Dec-2019

GENERAL COMMENTS	The paper addressed a very relevant study theme that has the potential to change the way clinicians respond to patients/bystanders in expressing morbidity related information in a pediatric cardiac surgery setting. Even though the theme is relevant, the methodological aspects show deficiency at several sections. The manuscript needs restructuring paying attention to methodological issues and interpretation. The writing style needs to be improved as well. Title 1. Title seems appropriate. Abstract 2. Abstract- the data was collected in 2014 as per the text. Why was it not published till now? Introduction 3. Introduction needs to be modified for better readability. The text appears long and would benefit from judicious editing. 4. There is no mention of the objectives of the study either in the introduction or in the methods section that follows it. Methodology 5. P5L46. In the participants and data collection section, the authors describe an online survey program. It fails to mention how the posted questions were decided by the investigators. Please provide a description of how the questions evolved, what revisions were made in the initial phase and what consultations were done to finalize this list. 6. P6L13. How was the focus group constructed? Was there any plan to include participants from different sections/roles of the society? The description suggests that it was constituted by responder self-selection and no attempt was made to include respondents from different sections/roles. This too will impair the generalizability of study findings. Also mention how many responded to the invitation to focus group discussion and how many were included. Results 7. P7L54. The sample of respondents shows a heavy bias towards mothers both in the online survey and the focus group discussion. Did the researchers do something to improve the response from fathers? The morbidity details and related factors are known to be perceived differently by fathers and mothers. The findings may be biased in this case as there is a definite selection bias in reference to gender. The sample also seems to be biased towards White/British population. These two are serious limitations that may impair generalizability of study findings. 8. The ability of the parents to understand communication from doctor depends on their education level and socioeconomic status
---

	also. Did your study look into these aspects before concluding on inability of parents to understand and comprehend the communication? If not please state that a limitation of the study. 9. The questionnaire includes (question number 1&2) parental perception of complication and morbidity. However, there is no discussion of the same in results. As a clinician one would like to know how the parents define complications/morbidity. Discussion 10. The discussion section needs to be modified for both structure and content. The introductory para of a discussion section is to summarize the salient points from results related to study objectives. The intermediary para are for discussing your main findings one by one relating them to other studies and interpreting differences/similarities. Please ensure that you close each discussion para with a summary line that distills the discussion in the para. 11. The discussion could be elaborated by suggesting remedies for the findings noted in the study. For eg a support staff like medical social worker/psychologist providing a continuous family support in non clinical aspects could alleviate the stress. 12. P14L23. In the limitation section it is stated that participants recalled conversations in a retrospective manner. Is there any data that can be added to the manuscript as to how far were these actual conversations/interactions that each respondent referred to in the survey/focus group discussion. An approximation of the recall period may add information to the possible bias from recall. Conclusion 13. P15L59. The conclusion section starts well and describes a summary of the findings as well as some suggestions for future. However, it is very long, less clear and loses focus at times. In addition, in this section, the authors quote some other studies and discuss/relates the study findings in this section. This may be moved to the discussion part as conventionally discussion relating to other studies are closed before conclusion. This change will also make the conclusion shorter and more direct. 14. P16L17. The para starting with “analysis...seems to be completely out of context here. This is sentence is incomplete to start with. Such text should definitely be in the discussion section above.
--	--

VERSION 2 – AUTHOR RESPONSE

Reviewer: 1

Dear Authors,

I have read the revised version of this paper.

You have made all the required corrections from both reviewers.

I still think it is worth asking for an extension on the world counting to be able to include into the body of the paper some of the quotations from parents.

I am now content with the paper and I think it is ready for publication.

Thank you for your review and comments.

Reviewer: 2

Dear authors,

The paper addressed a very relevant study theme that has the potential to change the way clinicians respond to patients/bystanders in expressing morbidity related information in a pediatric cardiac surgery setting. Even though the theme is relevant, the methodological aspects show deficiency at several sections. The manuscript needs restructuring paying attention to methodological issues and interpretation. The writing style needs to be improved as well.

Title

1. Title seems appropriate.

Abstract

2. Abstract- the data was collected in 2014 as per the text. Why was it not published till now?

The data were collected in 2014 as part of a wider study which was only recently completed. Whilst the data were analysed at the time of collection to inform subsequent stages of the project, competing pressures precluded earlier publication.

Introduction

3. Introduction needs to be modified for better readability. The text appears long and would benefit from judicious editing.

Thank you, we have taken this on board and edited the introduction significantly. Please see revised introduction.

4. There is no mention of the objectives of the study either in the introduction or in the methods section that follows it.

We have made this clearer in the introduction section, please see paragraph 2 of the introduction: "our objective was to elicit parents' perceptions about the way healthcare professionals communicate regarding morbidities associated with paediatric cardiac surgery."

Methodology

5. P5L46. In the participants and data collection section, the authors describe an online survey program. It fails to mention how the posted questions were decided by the investigators. Please provide a description of how the questions evolved, what revisions were made in the initial phase and what consultations were done to finalize this list.

We have added some additional text to explain how questions were developed.

6. P6L13. How was the focus group constructed? Was there any plan to include participants from different sections/roles of the society? The description suggests that it was constituted by responder self-selection and no attempt was made to include respondents from different sections/roles.

This too will impair the generalizability of study findings.

This is correct, the focus groups were made up of those who responded to the advertisements on the CHF website. The target audience was parents/carers from a broad ethnic and socio-economic mix, and this was borne in mind when the locations for the focus groups were chosen. Despite this, we did not achieve broad representation in the participants and this is discussed in the limitations section of the paper (page 15-16).

Also mention how many responded to the invitation to focus group discussion and how many were included.

All participants who wanted to attend were able to do so, although on the day not everyone did attend. We have added text to clarify this.

Results

7. P7L54. The sample of respondents shows a heavy bias towards mothers both in the online survey and the focus group discussion. Did the researchers do something to improve the response from fathers? The morbidity details and related factors are known to be perceived differently by fathers and mothers. The findings may be biased in this case as there is a definite selection bias in reference to gender. The sample also seems to be biased towards White/British population. These two are serious limitations that may impair generalizability of study findings.

Yes, we agree that this study unfortunately lacks the views of fathers and those of non-white British ethnicity, despite efforts to be inclusive with multiple methods of involvement (online forum vs focus groups) and focus group locations known for their ethnic diversity. We reflect on this in our limitations section of the paper and acknowledge that "in order to capture these important often unheard views, we must find and adopt an innovative approach that successfully includes minority groups to ensure the broadest capture of parental/family views".

8. The ability of the parents to understand communication from doctor depends on their education level and socioeconomic status also. Did your study look into these aspects before concluding on inability of parents to understand and comprehend the communication? If not please state that a limitation of the study.

We agree with the reviewer that ability to understand communication can be influenced by education level and socioeconomic status but in this study our focus was not on the ability of the parent/carer to understand, but rather on the ability of the clinician to communicate in a way that is understandable to the individual they are talking to. It was therefore not felt necessary or relevant to ask for information regarding the education level or socioeconomic status of our study participants.

9. The questionnaire includes (question number 1&2) parental perception of complication and morbidity. However, there is no discussion of the same in results. As a clinician one would like to know how the parents define complications/morbidity.

Yes, we agree that it is useful to know how parents define complications / morbidity. The responses to all the questions posed were analysed and found to sit in three broad themes, which we have outlined in tables 2 - 4 with illustrative quotes. For example, "Morbidity to me means death."

Discussion

10. The discussion section needs to be modified for both structure and content. The introductory para of a discussion section is to summarize the salient points from results related to study objectives.

The intermediary para are for discussing your main findings one by one relating them to other studies and interpreting differences/similarities. Please ensure that you close each discussion para with a summary line that distills the discussion in the para.

We have revised the discussion – and in particular the opening paragraphs – to improve the clarity and flow.

11. The discussion could be elaborated by suggesting remedies for the findings noted in the study. For eg a support staff like medical social worker/psychologist providing a continuous family support in non clinical aspects could alleviate the stress.

Thank you. We have added some further text about this into the discussion.

12. P14L23. In the limitation section it is stated that participants recalled conversations in a retrospective manner. Is there any data that can be added to the manuscript as to how far were these actual conversations/interactions that each respondent referred to in the survey/focus group discussion. An approximation of the recall period may add information to the possible bias from recall.

We do not have information about the time between conversations had and subsequent recall for discussions in the focus groups and forum. Our point here was that bias is present for any retrospective recall. We suggest prospective analysis of conversations in real time would provide valuable insights to the differences between what people say, what they think others understand from them, and what others actually understand.

Conclusion

13. P15L59. The conclusion section starts well and describes a summary of the findings as well as some suggestions for future. However, it is very long, less clear and loses focus at times. In addition, in this section, the authors quote some other studies and discuss/relates the study findings in this section. This may be moved to the discussion part as conventionally discussion relating to other studies are closed before conclusion. This change will also make the conclusion shorter and more direct.

We have significantly shortened the conclusion as suggested.

14. P16L17. The para starting with “analysis...seems to be completely out of context here. This is sentence is incomplete to start with. Such text should definitely be in the discussion section above.

Thank you for this feedback regarding the conclusion. We have edited this section considerably and hope you will find this a more succinct and focussed ending to the paper. Please see submitted revised conclusion for changes.

VERSION 3 – REVIEW

REVIEWER	Reviewer name: Shine Kumar Institution and Country: Amrita Institute of Medical sciences, Amrita Vishwa Vidyapeetham, India Competing interests: None
REVIEW RETURNED	16-Jan-2020
GENERAL COMMENTS	The revised manuscript is accepted for publication